# A Comparison for Infantile Mortality of Crucial Congenital Heart Defects in Korea over a Five-Year Period

**DOI:** 10.3390/jcm13216480

**Published:** 2024-10-29

**Authors:** Keesoo Ha, Chanmi Park, Junghwa Lee, Jeonghee Shin, Euikyung Choi, Miyoung Choi, Jimin Kim, Hongju Shin, Byungmin Choi, Soo-Jin Kim

**Affiliations:** 1Department of Pediatrics, College of Medicine, Korea University, Seoul 02841, Republic of Korea; kissuha@naver.com (K.H.); leejmd@korea.ac.kr (J.L.); sourer@hanmail.net (J.S.); ekchoi33@gmail.com (E.C.); cbmin@korea.ac.kr (B.C.); 2Biomedical Research Center, Korea University Guro Hospital, Seoul 08308, Republic of Korea; chanmipark@korea.ac.kr; 3National Evidence-Based Healthcare Collaborating Agency, Seoul 04933, Republic of Korea; mychoi@neca.re.kr (M.C.); jimin@neca.re.kr (J.K.); 4Department of Thoracic and Cardiovascular Surgery, Myoungju Hospital, Yongin 17050, Republic of Korea; babymedi@naver.com; 5Department of Pediatrics, Sejong General Hospital, Bucheon 14754, Republic of Korea

**Keywords:** congenital heart defect, mortality, critical illness, Korea

## Abstract

**Background**: Nearly half of congenital heart defects (CHDs) related to mortality occur during infancy although advancements in treatments have increased the survival rates. This study comprehensively examined overall and surgical mortality in CHD infants with the highest mortality rates in an effort to improve our understanding of CHD epidemiology. **Methods**: Participants were drawn from a dataset of 1,964,691 infants born between 2014 and 2018 in Korea. Crucial CHDs are defined here as including diverse categorical defects and classical critical CHDs but excluding simple shunt defects. Overall mortality (procedural and natural mortality) and procedural mortality (interventional and surgical mortality) for infants were analyzed. **Results**: The performance rate for multiple procedures in infants with crucial CHDs was 16%. The overall and surgical mortalities of crucial CHDs were 8% and 7%. The mortalities of palliative procedures were relatively high. Procedural mortalities for infants were significantly decreased in the tetralogy of Fallot (TOF), atrioventricular septal defects, and total anomalous pulmonary venous return (TAPVR) compared with overall mortalities for infants. Surgical mortalities for infants involving TOF and TAPVR were significantly lower, but those for infants involving hypoplastic left heart syndrome (HLHS) were higher than those for all ages. **Conclusions**: Palliative procedural techniques in infants must be improved to obtain better outcomes, particularly in the palliative surgery of HLHS. The infantile procedural outcomes for TOF and TAPVR are excellent and important in order to overcome disastrous circumstances during infancy. This comprehensive study of the overall and procedural mortalities of CHDs may have laid a cornerstone for CHD epidemiology in Korean infants.

## 1. Introduction

Congenital heart defects (CHDs) constitute the leading cause of infantile and neonatal mortality, and around 20–25% of life-threatening CHDs need urgent treatments within the period from birth to the first year of life [1]. Nearly half of all CHD-related mortality occurs during infancy although advances in diagnostic and therapeutic techniques have increased the survival rates of patients with CHDs [2].

South Korea’s National Health Insurance System (NHIS) has evolved dramatically since it was introduced in 1989 for all citizens. Interventional and surgical methods to treat CHDs are now covered by the NHIS, facilitating the identification of therapeutic outcomes for CHDs. Recently, Korean investigations reported that the surgical mortality rate between 2000 to 2014, from the data of the Korea Heart Foundation (KHF), was 8% in infants, and early and late surgical mortalities in infants’ data were 6.0% and 2.3%, respectively, according to KHF data [3,4]. However, most mortalities in Korea are reportedly associated with surgical procedures or restricted to specific populations. This study comprehensively examined the overall mortalities of CHDs regardless of surgical procedures and focused on infants who had had the highest mortality rates as part of an effort to better understand CHD epidemiology in Korean infants.

## 2. Material and Methods

### 2.1. Study Population

A total of 1,964,691 infants were born between 2014 and 2018 in South Korea. Crucial CHDs were defined as including diverse categorical defects [aortic stenosis (AoS), aortopulmonary window (APW), atrioventricular septal defect (AVSD), double-outlet left ventricle (DOLV), hypoplastic right heart syndrome (HRHS), pulmonary valve stenosis (PVS), and critical tricuspid stenosis (cTS)] and classical critical CHDs but excluding simple shunt defects [atrial septal defect (ASD), ventricular septal defect (VSD), and patent ductus arteriosus (PDA)]. Simple shunt defects including ASD, VSD, and PDA were excluded from this investigation because these defects, which constitute most CHDs, can distort the evaluation of mortality due to frequently duplicate diagnoses and procedures. The crucial CHDs were classified again according to five groups: cyanotic defects, double-outlet defects, left-to-right shunt, left-heart-obstructive defects, and right-heart-obstructive defects.

If complex CHDs had more than two diagnostic defects, they were classified simultaneously with multiple diagnoses. For example, in cases involving an atrioventricular septal defect (AVSD) with coarctation of aorta (CoA), AVSD could be one diagnosis and CoA could be the other diagnosis. If a subject had multiple diagnoses, all diagnostic names were included in this investigation in order to describe the numbers of cases and patients (a patient has multiple names of cases).

The representative tools for standardizing surgical procedures and categorizing risk stratification of complexity for CHDs were those established by the Society of Thoracic Surgeons—European Association for Cardio-Thoracic Surgery (STAT) and Risk Adjustment for Congenital Heart Surgery-1 (RACHS-1) [5]. The types of surgeries were assorted into STAT categories if possible. Mortality was analyzed according to age (neonate and infant), hospitalization (inpatient and outpatient), and procedure based on health insurance claims from the NHIS database in Korea. Mortality was classified according to following categories:

Category 1—inpatient mortality in those who had undergone procedures for CHDs and associated with the procedures.

Category 2—inpatient mortality in those who had undergone procedures for CHDs but not associated with the procedures.

Category 3—inpatient mortality in those who had not undergone a procedure for CHDs.

Category 4—outpatient mortality in those who had undergone procedures for CHDs.

Category 5—outpatient mortality in those who had not undergone a procedure for CHDs.

### 2.2. Data Source of Patients

Insurance claims based on the NHIS database were investigated in a retrospective and nationwide study. Target patients were searched and classified from the International Statistical Classification of Diseases and Related Health Problems, 10th Revision (ICD-10). Subjects (both inpatients and outpatients) comprised children from neonates to infants of one year after birth and the neonates were classified into preterm and term newborns. The investigation was performed from 2014 to 2018 for five years and yearly birth prevalence was calculated by neonatal live births, defined as the numbers of CHDs divided by 1000 neonatal live births.

Mortalities were classified as overall and procedural mortalities; overall mortality was defined as procedural and natural mortality was applied to individuals who had not undergone a medical procedure. Procedural mortality included both interventional and surgical mortality. The overall, procedural, surgical, and interventional mortalities were analyzed and compared with those reported previously.

Insurance claim data classified by the system of ICD-10 can be sorted into primary, secondary, and ruled-out diagnoses. Our approach of these detailed criteria focused on accurate identification of the CHD population and reducing the inclusion of false positives for more validity.

### 2.3. Ethical Statement

This study adhered to the country’s ethical guidelines for epidemiological studies and was approved by the ethics committee and Institutional Review Board (IRB) of the National Evidence-based Healthcare Collaborating Agency (IRB No. NECAIRB20-017-6). The need for informed consent was waived because the study design was secondary analysis and the participant data had been deidentified. The study was conducted in accordance with the principles of the Declaration of Helsinki (2013) and its foregoing amendment.

### 2.4. Statistical Analysis

Each mortality rate assigned to a subtype of crucial CHDs was analyzed and the mortalities were compared with those in other countries to increase the reliability of the analysis. The overall mortalities in Denmark, Norway, South Korea, and other nations in the review articles were compared with one another [6,7,8,9,10]. Surgical mortality rates reported by the KHF (Korea Heart Foundation) for all ages from 2000 to 2014 and our results (labeled Korean cardiac infants [KCIs] for infants from 2014 to 2018) were compared.

All data were presented as mean and standard deviation values and a statistically significant *p* value was considered as being less than 0.05. All statistical analyses were performed by SPSS version 20 program (IBM Corp., Armonk, NY, USA). Comparisons of categorical values were performed by means of χ^2^ or Fisher’s exact tests and comparisons of mean values were performed by means of a nonparametric Mann–Whitney U test.

## 3. Results

### 3.1. Baseline Characteristics

The number of crucial CHD patients was 4881 and the number of those cases was 5688 from 2014 to 2018 [11]. The yearly average numbers of CHD patients and cases were 976 and 1138, respectively. The birth prevalence of crucial CHDs per 1000 live births for patients was 2.5‰ and that for cases was 2.9‰. The preterm and full-term rates of crucial CHDs were 9% and 91%, respectively, and the most common frequencies of acyanotic and cyanotic CHDs were 24% for PVS and 17% for the tetralogy of Fallot (TOF) [11] (Table 1).

### 3.2. Overall and Procedural Mortalities

The number of overall mortalities was 395 out of 4881 CHDs, equivalent to 8%, and the infantile mortality rate was 20.10 among 100,000 live births (395 per 1,964,691). The procedural rate was 57% (with the intervention accounting for 10% and surgery for 47%) and the non-procedural rate was 43%. The procedural numbers for CHD patients and cases were 2786 and 3239, and the ratio of cases per patient (3239/2786) was 1.16, which was also the ratio of multiple procedures. The rate of procedural mortalities for CHD cases was 7% (213/3239), and the rates of interventional and surgical mortalities for CHD patients were 6% (33/627) and 7% (180/2655), respectively. The types of crucial CHDs with a high overall mortality were single ventricle (SV) (23%) and hypoplastic left heart syndrome (HLHS) (48%). Those with a low mortality were PVS (0.6%) and APW (4%). The types of crucial CHDs with a high interventional mortality were DORV (33%) and HLHS (60%), and those with a high surgical mortality were SV (13%) and HLHS (36%). The mortality values for overall mortality and procedural mortality showed significant differences in the TOF, AVSD, and TAPVR (*p* = 0.028, *p* = 0.012, and *p* < 0.001, respectively (Table 2).

### 3.3. Comparisons of Overall and Surgical Mortalities

The overall and surgical mortalities of infants according to the severities of CHDs were compared by country. The overall mortalities for all CHDs in Denmark and the review article were 5% and 4%, respectively [9,10]. The overall mortalities for severe CHDs (TOF, CoA, DORV, PuA, AVSD, transposition of the great arteries [TGA], TAPVR, AoS, Ebstein anomaly [EbA], HLHS, cTS, persistent truncus arteriosus [PTAr], interrupted aortic arch [IAA], and heterotaxia) in two Norwegian populations were 16% and 9% [8,9]. The overall mortality for crucial CHDs (TOF, CoA, DORV, PuA, AVSD, TGA, TAPVR, AoS, EbA, HLHS, HRHS/cTS, PTAr, IAA, SV, DOLV, APW, and PVS) in our KCIs was 8%. The overall mortality for non-severe CHDs (ASD, VSD, isolated PDA, PVS, minor valve malformation, and venous malformations) in the Norwegian population was 3% [8]. The overall mortality for crucial CHDs in the KCIs was significantly different from those for all CHDs in Denmark [9] and the review article (*p* < 0.001) [7], from those for severe CHDs in two Norwegian populations (*p* < 0.001 and *p* = 0.049, respectively) [8,9], and from that for non-severe CHDs in Norway (*p* < 0.001) [8] (Table 3).

The surgical mortality for severe CHDs in the Norwegian population, for crucial CHDs in the KCIs, and for non-severe CHDs in the Norwegian population were 8%, 8%, and 3%, respectively [8]. A 2023 study found that surgical mortality for all CHDs was approximately 3% [10]. The surgical mortality for crucial CHDs in the KCIs was significantly different from those for all CHDs in the review article (*p* < 0.001) [10] and for non-severe CHDs in Norway (*p* < 0.001) [8], but no significant difference was evident when compared with that for severe CHDs in Norway (*p* = 0.676) [8].

### 3.4. Individual Procedural Mortalities

The individual procedural mortalities were presented as number and percent values according to the subtypes of crucial CHDs and the procedures were classified into interventions and surgeries. The procedural mortalities according to the types of defects, and the defects’ mortalities according to the types of procedures, were presented as number and percent values. The individual procedural mortalities were presented by case mortality, not by patient mortality. The procedural, interventional, and surgical mortalities for individual cases were 7%, 5%, and 7%. The mortalities of palliative procedures were relatively high in PAS-B (17%), PAB (36%), AP shunt (16%), and Rp of HCx (29%) (Table 4).

### 3.5. Comparisons of Surgical Mortalities

The total and individual surgical mortalities of principal CHDs that only had raw data to be analyzed were compared between the KHF for all ages and the KCIs for infants. The total surgical mortality values in the two groups were 8% and 7%, respectively, and no statistical differences were evident in the total surgical mortality. However, the individual surgical mortality rates for HLHS, TOF, and TAPVR between the groups were significantly different (*p* = 0.032, 0.032, and <0.001, respectively) (Table 5).

### 3.6. Stratification of Individual Procedural Mortalities

The stratification of individual procedural mortality (interventional and surgical) in crucial CHDs was analyzed. Mortalities of palliative procedures were relatively high and high interventional mortalities (≥50%) included PAS-B for HLHS (69%) and PAB-B for DORV with TGA (50%), and high surgical mortalities (≥50%) included PAB of HLHS (55%) and PAB of SV (23%) (Table 6).

### 3.7. Characteristics of Mortalities

Mortality according to the subtypes of crucial CHDs was classified by age (neonate and infants), hospitalization (inpatient and outpatient), and procedure. The most common type of mortality was mortal category type 3 (inpatient mortality in patients who had not undergone a procedure for CHDs) in infants aged 1 to 12 months (except neonates), and the mortality was 128 (3%) among 4881 patients (Table 7).

## 4. Discussion

Multiple diagnoses could be estimated by the ratio of the number of patients and the number of cases. In other words, the cases were 5688 and the patients were 4881, the ratio of cases/patients was 1.2 (5688/4881) in our study, and therefore, the rate of multiple diagnoses was approximately 20% [11]. Many studies on the prevalence of CHDs have reported multiple diagnoses in a single CHD patient [12]. In addition, multiple procedures are also common in CHDs and should be assessed because a patient with a complex CHD can simultaneously or serially undergo various procedures. The rate of multiple procedures can be calculated as the ratio of cases per patient who has undergone procedures in the same way. The total procedural numbers of cases and patients were 3239 and 2786, respectively, and the ratio was 1.16 (3239/1786) for a 16% rate of multiple procedures.

Other studies have reported that in-hospital mortality in all types of CHDs is approximately 2–4% for all children with a CHD and 8–10% for newborns with a CHD [13,14]. A separate study reported that the total mortality rate of infants was 15–22 per 100,000 live births [15]. In our study, the overall, procedural, interventional, and surgical mortalities related to a CHD in infants were approximately 8%, 7%, 6%, and 7%, respectively, and the mortality rate per 100,000 live births was 20.10. The overall infantile mortality in our study was similar to those reported previously.

The mean mortalities of overall and surgical mortality in all crucial CHDs did not show significant difference (8% vs. 7%), but there were some differences in detailed types of CHDs (Table 2). The necessity and usefulness of procedural management were proven for infancy because the procedural mortalities were lower than the overall mortality including natural mortality in most crucial CHDs [10]. In particular, the procedural mortalities of the TOF, AVSD, and TAPVR in infants were significantly reduced when compared to the overall mortality of them. The procedural treatment in the TOF, AVSD, and TAPVR during infancy contributed to decrease procedural mortality, and therefore, procedural management in these defects during infancy is very important to enhance the survival rate.

Table 3 lists the differences in overall and surgical mortalities according to the severities of CHDs in infants in different countries. The overall mortality for all types of CHDs was 4–5%, and those for severe, crucial, and non-severe types of CHDs were 16–9%, 7%, and 3%, respectively [6,7,8,9]. Overall mortality gradually decreased with the severities of CHDs. This implies that the rate of overall mortality related to crucial CHDs is between those of severe and those of non-severe CHDs because crucial CHDs include severe CHDs and milder lesions of diverse categorical defects.

In spite of overall improvement in surgical outcomes, the mortalities of AP shunt and PA banding reported ranges from 2% to 16% and from 3% to 26% according to the underlying defects and conditions [16,17]. Our result showed relatively high mortalities in the palliative procedures of atrial septostomy, PA banding, AP shunt, and the repair of highly complicated defects (Rp of HCx) including Norwood operation. The palliative procedures may not represent a simple procedure and the correction of surgical techniques and risk stratification of defect needs for the improvement of surgical outcomes in Korea.

The success or failure of congenital cardiac surgery can be assessed by surgical mortality, which gradually decreased from 29% in 1975–1979 to 5% in 2005–2009 [18,19]. The Society of Thoracic Surgeons’ Congenital Heart Surgery Database for 2023 reported a 3% surgical mortality for all types of CHDs from 2019 to 2022 [10]. Using data from surgeries or interventions, the procedural mortality in infants varied from 4% to 12% according to age and procedural difficulty [20]. The surgical mortalities for severe, crucial, and non-severe CHDs were 8%, 8%, and 3%, respectively, in both Norwegian and Korean infants [8]. These results also show that surgical mortality gradually decreases by severity, although the difference between the surgical mortalities of crucial CHDs in our results and that of severe CHDs in Norway was not significant. Interventional mortality, which is rarely reported, for all CHDs in UK infants was 6% (28/472) [21]. This result was similar to the infant mortality for crucial CHDs in our study, which was 6% (33/627), indicating a nonsignificant difference between the two groups (*p* = 0.690).

The surgical mortality rates from the KHF for all ages and KCIs for infants are compared in Table 5 [3,4]. The surgical mortality rates associated with the TOF and TAPVR for infants are relatively low compared with those for all ages. Conversely, the surgical mortalities associated with HLHS for infants are relatively high compared with those for all ages.

### 4.1. TOF and TAPVR

Smith et al. reported that surgical mortality associated with in-hospital TOF was 2% for simple TOF and 4% for infants, but surgical mortalities in TAPVR cannot be regulated closely because TAPVR has a wide range of severities and subtypes [22,23]. In the KHF data, the surgical mortality rates associated with the TOF and TAPVR for all ages were 5% and 16% [3], and those for infants in the KCIs were 3% and 2%, respectively, and they showed significant differences. The surgical mortality rates for TOF and TAPVR during infancy were both significantly low although the surgical performances of the TOF and TAPVR in infancy (956 and 204) outnumbered those in all ages (679 and 86). This fact suggests that the outcome of surgical management for the TOF and TAPVR is excellent and important in order to overcome disastrous circumstances during infancy.

### 4.2. HLHS

Cervantes-Salazar revealed that the surgical mortality rates of HLHS at 1 month and 1 year of life were 28–47% and 12–29%, respectively [24]. The surgical mortality of HLHS for all ages according to the KHF was 20% [3] while that for infants in the KCIs was as high as 36%. The surgical mortality of HLHS in the KCIs for infants was relatively high compared with that for all ages but it may not be high compared with other national studies. It has been reported that left-heart-obstructive defects including HLHS are less common in the East compared to the West [14]. The possibility that the surgical mortality rate is high in the East cannot be ruled out due to the relatively low amount of surgical experience such as with the Norwood operation. Also, high surgical mortality rates of HLHS during infancy can be associated with high surgical mortality for palliation because our results mentioned high surgical mortalities in palliative surgeries such as PA banding and AP shunt, which are associated with the palliation of HLHS. Palliative surgical techniques of HLHS in infants must be improved to obtain better surgical outcomes although this defect has wide spectra of severity and heterogeneity.

The stratification of the TOF, TAPVR, and HLHS and age groups could be improved by offering additional insights into the factors that influence the observed mortality, and additional investigation should be performed.

### 4.3. AVSD, D-TGA, CoA, and PuA

The surgical mortality rate associated with AVSD, D-TGA, CoA, and PuA varies by disease complexities and individual subtypes. Devaney and colleagues reported that the surgical mortality of AVSD was 2–6% and a late mortality was 6% at a median follow-up of 5.5 years [25]. Planche et al. reported that early mortality for D-TGA was 9% and that for late mortality was 2% [26]. St Louis et al. revealed that surgical mortality associated with CoA was greatest within the first week of life, at 10%, but this decreased to less than 3% with age [27]. Grant et. al. reported surgical mortality rates of PuA with an intact ventricular septum (IVS) of 7% for all ages and 11% for neonates and infants [28]. Kwak et al. and Babliak et al. found that the surgical mortality for PuA with VSD was 9% for up to an age of 90 days, and the cumulative mortality was 4% for patients with a median age of 5 years [29,30]. Surgical mortalities associated with PuA were compared with one another as a sum of PA with VSD and IVS because the NHIS data did not differentiate between the defects.

Surgical mortalities for AVSD, D-TGA, CoA, and PuA using the KHF for all ages did not show significant differences compared with those using the KCIs for infants [3]. The complexity of subtypes and risk stratification of surgeries for these defects cannot be easily classified, and detailed classification will be needed in future studies.

This study had some limitations. First, the therapeutic methods used to treat CHDs, particularly surgeries, differed according to the severity of disease, even with the same CHD diagnosis. Therefore, some interventions or surgeries that are not commonly used from among various therapeutic methods can be omitted in the process of finding out optimal procedures for specific diseases. Second, the classification of surgery for complicated CHDs was not subdivided in detail but was classified as the repair of highly complicated (Rp HCx) defects in the Korean application of the ICD-10 system. For example, Norwood operations and Damus–Kaye–Stansel procedures in Categories 6 or 5 of RACHS-1 or STAT classifications, which standardize the risk stratification of surgery, were not classified separately but integrated into a single group as highly complicated (Rp HCx) defects in Korean ICD-10. We were unable to associate them with STAT and RACHS-1 categories.

## 5. Conclusions

The comprehensive palliative procedures showed relatively high mortalities and the improvement of surgical techniques in palliative procedure needs for better surgical outcomes in Korea. The procedural mortalities in most crucial CHDs were lower than the overall mortality during infancy. Particularly, the procedural mortalities of the TOF, AVSD, and TAPVR in infants were significantly reduced when compared to those of the overall mortality and procedural treatment in these CHDs during infancy is very important. Also, the surgical mortality of the TOF and TAPVR for infants was significantly low compared with that for all ages. The infantile surgical outcomes for the TOF and TAPVR are excellent and important in order to overcome disastrous circumstances during infancy. Conversely, the surgical mortality of HLHS in infants showed a significant increase when compared to that in children of all ages, and a high surgical mortality of HLHS during infancy can be associated with high palliative mortality. Careful consideration is needed to avoid missing therapeutic methods that are most suitable for treating the disease but are not commonly used. Future studies will need to expand to other topics such as inherited arrhythmias and the mortality of CHDs and to construct detailed systems of Korea’s ICD-10 for the stratification of severity in complex CHDs. This comprehensive study of overall and procedural mortalities of CHDs for infants may have laid a cornerstone for an improved understanding of CHD epidemiology in Korean infants despite the limitations of ICD-10 classification.

## Figures and Tables

**Table 1 jcm-13-06480-t001:** The number (‰) of birth prevalences for crucial congenital heart defects (CHDs) in Korean infants from 2014 to 2018 [11].

Fr	Defect	Five-Year Mean No. (%)	Full-Term No.	Birth Prevalence No. (‰)	Fr	Defect	Five-Year Mean No. (%)	Full-Term No.	Birth Prevalence No. (‰)
	**Total Patients**	**976**	**4421**	**4881 (2.5‰)**		**Total Cases**	**1138**	**5159**	**5688 (2.9‰)**
1	PVS	273 (24)	1255	1365 (0.69‰)	7	D-TGA	71 (6)	333	355 (0.18‰)
2	TOF	188 (17)	841	938 (0.48‰)	8	TAPVR	47 (4)	222	235 (0.12‰)
3	CoA	124 (11)	552	618 (0.31‰)	9	SV	40 (4)	192	201 (0.10‰)
4	DORV	107 (9)	486	534 (0.27‰)	10	AoS ^†^	37 (3)	161	184 (0.09‰)
	DORV-VSD	89 (8)	400	444 (0.23‰)	11	EbA	23 (2)	96	115 (0.06‰)
	DORV-PS	10 (1)	47	50 (0.03‰)	12	HLHS	19 (2)	83	94 (0.05‰)
	DORV-TGA	8 (1)	39	40 (0.02‰)	13	HRHS/cTS	14 (1)	63	68 (0.03‰)
5	PuA *	87 (8)	373	437 (0.22‰)	14	APW	10 (1)	45	52 (0.03‰)
6	AVSD	75 (7)	349	376 (0.19‰)	15	L-TGA	10 (1)	47	50 (0.03‰)
	AVSD-simple	70 (6)	323	349 (0.18‰)	16	PTAr	9 (1)	40	45 (0.02‰)
	AVSD-CoA	4 (0.4)	20	20 (0.01‰)	17	DOLV	3 (0.3)	13	13 (0.01‰)
	AVSD-PS	1 (0.1)	6	7 (0.00‰)	18	IAA	2 (0.2)	8	8 (0.004‰)

* PuA (pulmonary atresia) includes atresia of pulmonary artery and valve; ^†^ AoS (aortic stenosis) includes stenosis of aortic artery and valve. Abbreviations—AoS: aortic stenosis, APW: aortopulmonary window, AVSD: atrioventricular septal defect, CoA: coarctation of aorta, cTS: critical tricuspid stenosis, DOLV: double-outlet left ventricle, DORV: double-outlet right ventricle, EbA: Ebstein anomaly, Fr: frequency, HLHS: hypoplastic left heart syndrome, HRHS: hypoplastic right heart syndrome, IAA: interrupted aortic arch, PS: pulmonary stenosis, PuA: pulmonary atresia, PVS: pulmonary valve stenosis, PTAr: persistent truncus arteriosus, SV: single ventricle, TAPVR: total anomalous pulmonary venous return, TGA: transposition of aorta, TOF: tetralogy of Fallot, and VSD: ventricular septal defect.

**Table 2 jcm-13-06480-t002:** Overall and procedural (interventional and surgical) mortalities of crucial CHDs in Korean infants over a five-year period.

Defect	Overall CHD			Procedure (57%)		Intervention (10%)		Surgery (47%)		*p* (Mortality)
	**No.**	**Mortality (%)**	**Mortality Rate ^‡^**	**No.**	**Mortality (%)**	**No.**	**Mortality (%)**	**No.**	**Mortality (%)**	**Overall vs.**
	**4881**	**395 (8)**	**20.10**	**3239**	**213 (7)**	**627**	**33 (6)**	**2655**	**180 (7)**	**Procedural**
PVS	1365	7 (0.6)	0.36	224	1 (0.4)	219	1 (0.5)	5	0	0.686
TOF	938	45 (6)	2.29	1128	33 (3)	172	2 (1)	956	31 (3)	0.028
CoA	618	30 (6)	1.53	370	20 (5)	38	3 (8)	339	17 (5)	0.765
DORV	534	43 (9)	2.19	56	2 (4)	3	1 (33)	53	1 (2)	0.298
PuA *	437	48 (13)	2.44	325	33 (10)	45	1 (2)	280	32 (11)	0.812
AVSD	376	38 (12)	1.93	177	7 (4)			177	7 (4)	0.012
D-TGA	355	27 (9)	1.37	368	30 (8)	110	12 (10)	258	18 (7)	0.890
TAPVR	235	40 (20)	2.04	206	4 (2)	2	0	204	4 (2)	<0.001
SV	201	39 (23)	1.99	205	26 (13)			205	26 (13)	0.078
AoS ^†^	184	15 (10)	0.76	30	5 (17)	23	4 (17)	7	1 (14)	0.169
EbA	115	10 (10)	0.51							
HLHS	94	39 (48)	1.99	131	51 (39)	15	9 (60)	118	42 (36)	0.783
HRHS/cTS	68	3 (5)	0.15	12				36	0	
APW	52	2 (4)	0.10							
L-TGA	50	3 (7)	0.15	5				5	0	
PTAr	45	4 (10)	0.20							
DOLV	13	2 (18)	0.10	7	1 (14)			7	1 (14)	0.730
IAA	8	0 (0)	0	5				5	0	

* PuA (pulmonary atresia) includes atresia of pulmonary artery and valve. ^†^ AoS (aortic stenosis) includes stenosis of aortic artery and valve. ^‡^ Mortality rate for infants is defined as number of deaths in infants per 100,000 live births during five years in Korea. Abbreviations—AoS: aortic stenosis, APW: aortopulmonary window, AVSD: atrioventricular septal defect, CoA: coarctation of aorta, cTS: critical tricuspid stenosis, DOLV: double-outlet left ventricle, DORV: double-outlet right ventricle, EbA: Ebstein anomaly, HLHS: hypoplastic left heart syndrome, HRHS: hypoplastic right heart syndrome, IAA: interrupted aortic arch, PuA: pulmonary atresia, PVS: pulmonary valve stenosis, PTAr: persistent truncus arteriosus, SV: single ventricle, TAPVR: total anomalous pulmonary venous return, TGA: transposition of aorta, TOF: tetralogy of Fallot, and VSD: ventricular septal defect.

**Table 3 jcm-13-06480-t003:** Comparisons of overall and surgical mortalities according to severity of CHDs in infants by country.

Nation	CHD			Surgery		Duration (Years)
	Severity	No.	Overall Mortality	Eligible No.	Mortality	
Denmark ^(6)^	All	3365	5% (168/3365)	2416 (72%)	-	13 (2003–2015)
Review articles ^(7,10)^	All ^(7)^	23,651	4% (986/23,651)	91,798 ^(10)^	3% (2464/91,798)	4 (2019–2022)
Norway ^(8)^	Severe *	2673	16% (425/2673)	1642 (61%)	8% (138/1642)	16 (1994–2009)
Norway ^(9)^	Severe *	2359	9% (219/2359)	-	-	-
Korea	Crucial ^†^	4881	8% (395/4881)	2312 (47%)	8% (180/2312)	5 (2014–2018)
Norway ^(8)^	Non-severe ^‡^	8599	3% (254/8599)	631 (7%)	3% (20/631)	16 (1994–2009)

* Severe CHDs in Norway: TOF, CoA, DORV, PuA, AVSD, TGA, TAPVR, AoS, EbA, HLHS, cTS, PTAr, IAA, heterotaxia, and others. ^†^ Crucial CHDs in Korea: TOF, CoA, DORV, PuA, AVSD, TGA, TAPVR, AoS, EbA, HLHS, HRHS/cTS, PTAr, IAA, SV, DOLV, APW, and PVS. ^‡^ Non-severe CHDs in Norway: ASD, VSD, isolated PDA, PVS, minor valve malformation, venous malformations, and others. ^(6–10)^ present the number of references.

**Table 4 jcm-13-06480-t004:** The number of individual procedural mortalities of crucial CHDs in Korean infants over a five-year period.

Procedure	Case Mortality	LR Shunt	DO Defect	Cyanotic Defect				Left- and Right-Heart-Obstructive Defect				
	**% (n)**	**AVSD**	**DORV ^(1)^ DOLV ^(2)^**	**TOF**	**D-TGA ^(3)^** **L-TGA ^(4)^**	**TAPVR**	**SV**	**CoA**	**AoS ^(5)^ *** **IAA ^(6)^**	**HLHS ^(7)^** **HRHS/cTS ^(8)^**	**PVS**	**PuA ^†^ **
Total	7% (213/3230)	4 (7/177)	4 (2/56) ^(1)^ 33 (1/3) **^(^**^2)^	3 (33/1128)	8 (30/358) ^(3)^ 0 (0/2) ^(4)^	2 (4/206)	13 (26/205)	5 (20/370)	17 (5/30) ^(5)^ 0 (0/3) ^6)^	39 (51/131) ^(7)^ 0 (0/12) ^8)^	0.5 (1/224)	11 (33/325)
Intervention	5% (30/619)											
PPV/PAV	2 (6/331)		0 (0/1) ^(1)^	1 (1/80)				0 (0/1)	18 (3/17) ^5)^		0.5 (1/205)	4 (1/27)
PTA-P/PTA-A	3 (5/167)			1 (1/92)				8 (3/37)	17 (1/6) ^5)^		0 (0/14)	0 (0/18)
PAS-B	17 (19/111)		50 (1/2) ^(1)^		10 (9/94) ^(3)^	0 (0/2)				69 (9/13) ^(7)^		
PAS-K	0 (0/10)				0 (0/10) ^(3)^							
Surgery (STAT)	7% (180/2611)											
AS/PS op	11 (2/19)							14 (1/7)	14 (1/7) ^(5)^		0 (0/5)	
PAB (4)	36 (35/97)						23 (12/52)		0 (0/3) ^(6)^	55 (23/42) ^(7)^		
AP shunt (4)	16 (50/315)		0 (0/10) ^(1)^	7 (9/122)			23 (11/48)			47 (7/15) ^(7)^		19 (23/120)
Rastelli (3)	4 (2/49)											4 (2/49)
Glenn (2)	2 (3/182)		0 (0/2) ^(2)^				3 (3/105)			0 (0/33) ^(7)^ 0 (0/12) ^8)^		0 (0/30)
Cr TOF (2–3); TAPVR; CoA (1–3)	2 (13/642); 2 (4/204); 5 (16/334)	0 (0/9)	0 (0/4) ^(1)^	2 (13/638)		2 (4/204)		5 (16/325)				
LR PAR	5 (9/189)			3 (5/157)								13 (4/32)
Rp TGA (3–4)	7 (19/257)		1 (1/10) ^(1)^		7 (18/245) ^(3)^ 0 (0/2) ^(4)^							
Rp Cx (3–5) ^‡^	14 (8/59)		0 (0/29) ^(1)^		0 (0/4) ^(3)^					42 (8/19) ^(7)^		0 (0/7)
Rp HCx (3–5) ^§^	29 (4/14)				0 (0/5) ^(3)^					44 (4/9) ^(7)^		
LV or RV OTR	11 (9/83)	100(1/1)	100 (1/1) ^(2)^	10 (4/39)								7 (3/42)
Cr pAVSD	5 (2/44)	5 (2/44)										
Cr cAVSD (3)	3 (4/123)	3 (4/123)										

* PuA (pulmonary atresia) includes atresia of pulmonary artery and valve. ^†^ AoS (aortic stenosis) includes stenosis of aortic artery and valve. ^‡^ Customary types of repair of complicated CHDs in Korea: arch hypoplasia correction via sternotomy approach, repair of Ebstein anomaly, DKS (Damus-Kaye-Stansel) procedure. ^§^ Customary types of repair of highly complicated CHDs in Korea: Norwood operation, Yasui (Norwood-Rastelli) operation, aortic translocation. Abbreviations of diagnosis (AoS: aortic stenosis, AVSD: atrioventricular septal defect, CoA: coarctation of aorta, cTS: critical tricuspid stenosis, DOLV: double-outlet left ventricle, DORV: double-outlet right ventricle, HLHS: hypoplastic left heart syndrome, HRHS: hypoplastic right heart syndrome, IAA: interrupted aortic arch, PS: pulmonary stenosis, PuA: pulmonary atresia, PVS: pulmonary valve stenosis, SV: single ventricle, TAPVR: total anomalous pulmonary venous return, TGA: transposition of aorta, and TOF: tetralogy of Fallot. Abbreviations of procedures and surgeries—Cr: correction, Cx: complicated, HCx: highly complicated, PAB: pulmonary artery banding, LR PAR: left and right pulmonary artery reconstruction, PAS-B: percutaneous atrial septostomy-balloon, PAS-K: percutaneous atrial septostomy-knife, PAV: percutaneous aortic valvuloplasty, PPV: percutaneous pulmonary valvuloplasty, PTA-A: percutaneous transluminal angioplasty—aortic, PTA-P: percutaneous transluminal angioplasty—pulmonary, Rp: repair, and LV or RV OTR: left ventricular or right ventricular outflow tract reconstruction. ^(1–8)^ indicate each defect to distinguish defects in the same column (DORV vs. DORV, D-TGA vs. L-TGA, AoS vs. IAA, and HLHS vs. HRHS cTS).

**Table 5 jcm-13-06480-t005:** Comparisons of surgical mortalities of principal CHDs between the Korea Heart Foundation (KHF) and Korean cardiac infants (KCIs) of our investigation.

**Population/Year**	**All Ages/2000–2014 (KHF)**						**Infants/2014–2018 (KCIs)**					
**Total Surgical Mortality**	**8% (214/2617)**						**7% (180/2655)**					
**Subgroup**	**Left-Heart-Obstructive Defect**						**Right-Heart-Obstructive Defect**				**LR Shunt**	
**Diagnosis**	**CoA**		**AoS**		**HLHS ***		**PuA (VSD and IVS)**		**PVS**		**AVSD**	
**Year**	**KHF**	**KCIs**	**KHF**	**KCIs**	**KHF**	**KCIs**	**KHF**	**KCIs**	**KHF**	**KCIs**	**KHF**	**KCIs**
Each surgical mortality	6% (12/205)	5% (17/339)	17% (1/6)	14% (1/7)	20% (14/70)	36% (42/118)	8% (40/475)	12% (32/280)	3% (2/74)	0% (0/5)	8% (18/233)	4% (7/177)
**Subgroup**	**Cyanotic defect**										**Double-outlet defect**	
**Diagnosis**	**TOF ***		**D-TGA**		**L-TGA**		**TAPVR ***		**SV**		**DORV**	
**Year**	**KHF**	**KCIs**	**KHF**	**KCIs**	**KHF**	**KCIs**	**KHF**	**KCIs**	**KHF**	**KCIs**	**KHF**	**KCIs**
Each surgical mortality	5% (37/679)	3% (31/956)	8% (13/157)	7% (18/258)	22% (16/73)	0% (0/5)	16% (14/86)	2% (4/204)	11% (67/589)	13% (26/205)	9% (19/212)	2% (1/53)

* The subtypes of CHDs with *p* < 0.05 between the KHF and KCIs (*p* = 0.032 at HLHS, *p* = 0.032 at TOF, and <0.001 at TAPVR). Abbreviations—AoS: aortic stenosis, AVSD: atrioventricular septal defect, CoA: coarctation of aorta, DORV: double-outlet right ventricle, HLHS: hypoplastic left heart syndrome, PuA: pulmonary atresia, PVS: pulmonary valve stenosis, SV: single ventricle, TAPVR: total anomalous pulmonary venous return, TGA: transposition of aorta, TOF: tetralogy of Fallot, and VSD: ventricular septal defect.

**Table 6 jcm-13-06480-t006:** The stratification of individual procedural mortalities of crucial CHDs in Korean infants over a five-year period.

Mortality	Intervention (%)	Surgery (%)
≥50%	PAS-B of HLHS (69), PAS-B of DORV-TGA (50)	PAB of HLHS (55)
20–50%		PAB of SV (23)
		AP shunt of SV (23), AP shunt of HLHS (47)
		Rp Cx of HLHS (42), Rp HCx of HLHS (44)
10–20%	PAV of AoS (18), PTA-A of AoS (17)	AP shunt of PuA (19)
	PAS-B of D-TGA (13)	LR PAR of PuA (13)
		RVOTR of TOF (10)
		Rp TGA of DORV-TGA (10)

Abbreviations of diagnosis—AoS: aortic stenosis, AVSD: atrioventricular septal defect, DORV: double-outlet right ventricle, PuA: pulmonary atresia, SV: single ventricle, TGA: transposition of aorta, and TOF: tetralogy of Fallot. Abbreviations of procedures and surgeries—Cx: complicated, HCx: highly complicated, PAB: pulmonary artery banding, LR PAR: left and right pulmonary artery reconstruction, PAS-B/K: percutaneous atrial septostomy-balloon/knife, PAV: percutaneous aortic valvuloplasty, PTA-A: percutaneous transluminal angioplasty—aortic, Rp: repair, and RVOTR: right ventricular outflow tract reconstruction.

**Table 7 jcm-13-06480-t007:** The characteristics of mortalities according to subtypes of crucial CHDs in Korean infants over a five-year period.

Type	Patient	Mortality (%)										
	**Fr**	**Total**	**Category 1 ***		**Category 2 ^†^**		**Category 3 ^‡^**		**Category 4 ^§^**		**Category 5 ^‖^**	
			**≤1 m**	**1 to 12 m**	**≤1 m**	**1 to 12 m**	**≤1 m**	**1 to 12 m**	**≤1 m**	**1 to 12 m**	**≤1 m**	**1 to 12 m**
	**4881**	**395 (8)**	**43 (1)**	**106 (2)**		**44 (1)**	**36 (1)**	**128 (3)**		**21 (0.4)**		**17 (0.3)**
PVS	1365	7 (0.5)	1 (0.1)					4 (0.3)		1 (0.1)		1 (0.1)
TOF	938	45 (5)	4 (0.5)	13 (1)		6 (1)	3 (0.5)	12 (1)		3 (0.5)		4 (0.5)
CoA	618	30 (5)	5 (1)	9 (1)		3 (0.5)	4 (1)	5 (1)		2 (0.3)		2 (0.3)
DORV	534	43 (8)		3 (0.6)			4 (0.7)	35 (6.6)				1 (0.2)
DORV-VSD	444	40 (9)		1 (0.2)			4(1)	34 (7.5)				1 (0.2)
DORV-PS	50	1 (2)						1 (2)				
DORV-TGA	40	2 (5)		2 (5)								
PuA	437	48 (11)	5 (1)	14 (4)		9 (2)	5 (1)	9 (2)		5 (1)		1 (0.2)
AVSD	376	38 (10)		2 (0.5)		5 (1)	6 (2)	24 (6)		1 (0.3)		
AVSD-simple	349	34 (10)		2 (1)		4 (1)	4 (1)	23 (7)		1 (0.3)		
AVSD-CoA	20	3 (15)				1 (5)	2 (10)					
AVSD-PS	7	1 (14)						1 (14)				
D-TGA	355	27 (8)	6 (1)	13 (3.5)		2 (0.5)		4 (1)		1 (1)		1 (1)
TAPVR	235	40 (17)	9 (4)	20 (8)		7 (3)		4 (2)				
SV	201	39 (19)	4 (2)	11 (5)		9 (5)	3 (1)	7 (4)		5 (2)		
AoS	184	15 (8)	1 (0.5)	2 (1)		2 (1)	1 (0.5)	8 (4.5)				1 (0.5)
EbA	115	10 (9)					4 (4)	5 (6)				1 (1)
HLHS	94	39 (41)	8 (9)	19 (20)		1 (1)	2 (2)	4 (4)		2 (2)		3 (3)
HRHS/cTS	50	3 (6)					1 (2)	1 (2)				1 (2)
APW	68	2 (3)					1 (1.5)	1 (1.5)				
L-TGA	52	3 (6)						2 (4)				1 (2)
PTAr	4	4 (9)					2 (4.5)	2 (4.5)				
DOLV	13	2 (15)						1 (7.5)		1 (7.5)		
IAA	8											

* Category 1—inpatient mortality in those who had undergone procedures for CHDs and associated with the procedures. ^†^ Category 2—inpatient mortality in those who had undergone procedures for CHDs but not associated with the procedures. ^‡^ Category 3—inpatient mortality in those who had not undergone a procedure for CHDs. ^§^ Category 4—outpatient mortality in those who had undergone procedures for CHDs. ^‖^ Category 5—outpatient mortality in those who had not undergone a procedure for CHDs. Abbreviations—AoS: aortic stenosis, APW: aortopulmonary window, AVSD: atrioventricular septal defect, CoA: coarctation of aorta, cTS: critical tricuspid stenosis, DOLV: double-outlet left ventricle, DORV: double-outlet right ventricle, EbA: Ebstein anomaly, Fr: frequency, HLHS: hypoplastic left heart syndrome, HRHS: hypoplastic right heart syndrome, IAA: interrupted aortic arch, PS: pulmonary stenosis, PuA: pulmonary atresia, PVS: pulmonary valve stenosis, PTAr: persistent truncus arteriosus, SV: single ventricle, TAPVR: total anomalous pulmonary venous return, TGA: transposition of aorta, TOF: tetralogy of Fallot, and VSD: ventricular septal defect.

## Data Availability

Data are available from the authors upon reasonable request with the permission of the Institutional Review Board of the National Evidence-based Healthcare Collaborating Agency. The data associated with this article can be obtained from the corresponding authors upon reasonable request.

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
