# Peer review of "A Comparison for Infantile Mortality of Crucial Congenital Heart Defects in Korea over a Five-Year Period"

_jcm, 2024, doi:10.3390/jcm13216480_

Round 1
Reviewer 1 Report
Comments and Suggestions for Authors
The article “A Comparison for Infantile Mortality of Crucial Congenital Heart Defects in Korea over a Five-year Period” is well written and presents an important topic related to infantile mortality of congenital heart disease in Korea over a five-year period.
The introduction, methods and references seem appropriately described.
However, the discussion and conclusion sections need some more depth. It would be nice to tie the results towards the discussion.
· Some of the sentences need some rewording, for example, this sentence does not convey a clear message, were there differences or not: Page 11 “line 311- 312, The mortalities between overall mortality and surgical mortality did not show difference (8% vs. 7%), but there were little differences in detailed items”.
· The next few lines where it mentions that the necessity and usefulness of procedural management has been proven during infancy because the procedural mortalities were lower than the overall mortality including natural mortality in most crucial CHDs, also needs some more, explaining with references and are these results explained before.
· Limitations and conclusions also would merit from some more content. In the limitations section, future directions to overcome these limitations by looking into the surgical or various therapeutic methods that can be omitted in the process of finding out optimal procedures for specific diseases, as well.
Author Response
Comments and Suggestions for Authors: Reviewer 1
Thank you for pointing this out. We agree with this comment. Therefore, we have changed my manuscript according to your comments.
The article “A Comparison for Infantile Mortality of Crucial Congenital Heart Defects in Korea over a Five-year Period” is well written and presents an important topic related to infantile mortality of congenital heart disease in Korea over a five-year period.
The introduction, methods and references seem appropriately described.
However, the discussion and conclusion sections need some more depth. It would be nice to tie the results towards the discussion.
- Some of the sentences need some rewording, for example, this sentence does not convey a clear message, were there differences or not: Page 11 “line 311- 312, The mortalities between overall mortality and surgical mortality did not show difference (8% vs. 7%), but there were little differences in detailed items”. à Mean mortalities of overall and surgical mortality in all crucial CHDs did not show significant difference (8% vs. 7%), but there were some differences in detailed types of CHDs (Table 2). [Page 11, line 312- 314]
- The next few lines where it mentions that the necessity and usefulness of procedural management has been proven during infancy because the procedural mortalities were lower than the overall mortality including natural mortality in most crucial CHDs, also needs some more, explaining with references and are these results explained before. à The necessity and usefulness of procedural management has been proven during infancy because the procedural mortalities were lower than the overall mortality including natural mortality in most crucial CHDs [10]. [Page 11, line 314- 316]
- Limitations and conclusions also would merit from some more content. In the limitations section, future directions to overcome these limitations by looking into the surgical or various therapeutic methods that can be omitted in the process of finding out optimal procedures for specific diseases, as well. à Careful consideration is needed to avoid missing therapeutic methods that are most suitable for treating the disease but are not commonly used.
[conclusions Page 13, line 420- 422]

Reviewer 2 Report
Comments and Suggestions for Authors
The explanation of the target patient group, which includes "crucial CHDs" and excludes simpler conditions like VSD and ASD, could be improved by providing more detail on the rationale for excluding certain conditions.
While the definitions of procedural and natural mortality are clear, there could be more elaboration on how non-procedural mortality cases (Category 3 and Category 5) were identified and classified, to clarify any potential concerns about misclassification or unreported deaths.
Providing a more detailed explanation of the confidence intervals and effect sizes would strengthen the statistical analysis and improve its rigor.
The stratification of specific defects (e.g., HLHS, TOF, TAPVR) and age groups could be improved by offering additional insights into the factors that influenced the observed mortality rates.
The discussion could be enriched by including more comparisons with global data on CHD mortality and surgical outcomes. Moreover to give the manuscript a broader scope authors should include and briefly discuss not only CHD but also inherited Arrhythmogenic syndromes (Inherited Arrhythmias in the Pediatric Population: An Updated Overview. Medicina (Kaunas). 2024 Jan 3;60(1):94. doi: 10.3390/medicina60010094. )
The limitations are clearly outlined; however, it would be helpful to discuss ways in which some of these could be addressed in future research, particularly regarding the challenges posed by ICD-10 coding in accurately classifying complex CHDs.
Author Response
Comments and Suggestions for Authors: Reviewer 2
Thank you for pointing this out. We agree with this comment. Therefore, we have changed my manuscript according to your comments.
The explanation of the target patient group, which includes "crucial CHDs" and excludes simpler conditions like VSD and ASD, could be improved by providing more detail on the rationale for excluding certain conditions. à Simple shunt defects of ASD, VSD, and PDA were excluded from this study because they which comprise most CHDs can distort the evaluation of mortality due to frequently duplicate diagnoses and procedures. [Page 2, line 60- 62]
While the definitions of procedural and natural mortality are clear, there could be more elaboration on how non-procedural mortality cases (Category 3 and Category 5) were identified and classified, to clarify any potential concerns about misclassification or unreported deaths. à [Explanation: Based on Korea's insurance claims data, mortality related to outpatient and inpatients (hospitalization), interventions and surgeries can be easily confirmed.] à based on health insurance claims from the NHIS database in Korea. [Page 2, line 76- 77]
Providing a more detailed explanation of the confidence intervals and effect sizes would strengthen the statistical analysis and improve its rigor. à [Explanation: The resulting value of a country's mortality rate for each year is presented as a single statistical number, so it cannot be displayed as an average or confidence interval, but is presented as a single value.]
The stratification of specific defects (e.g., HLHS, TOF, TAPVR) and age groups could be improved by offering additional insights into the factors that influenced the observed mortality rates. à [ I appreciate your comment and I added your comment with revised forms.] à The stratification of TOF, TAPVR, and HLHS and age groups could be improved by offering additional insights into the factors that influenced the observed mortality, and additional investigation should be performed. [Page 12, line 377- 379]
The discussion could be enriched by including more comparisons with global data on CHD mortality and surgical outcomes. Moreover, to give the manuscript a broader scope authors should include and briefly discuss not only CHD but also inherited Arrhythmogenic syndromes (Inherited Arrhythmias in the Pediatric Population: An Updated Overview. Medicina (Kaunas). 2024 Jan 3;60(1):94. doi: 10.3390/medicina60010094.) à [Explanation: Thank you for your valuable opinion. However, this study did not include research on inherited arrhythmia, so it could not be compared with studies in other countries. A statement was attached to consider research on inherited arrhythmia in future plans. à Future researches will need to expand to other topics such as inherited arrhythmias as well as the mortality of CHDs, [Page 13, line 422-423]
The limitations are clearly outlined; however, it would be helpful to discuss ways in which some of these could be addressed in future research, particularly regarding the challenges posed by ICD-10 coding in accurately classifying complex CHDs. à Future researches will need ~ and to construct detailed system of Korea’s ICD-10 for stratification of severity in complex CHDs. [Page 13, line 423-424]

Round 2
Reviewer 2 Report
Comments and Suggestions for Authors
I must congratulate to the authors for having answered to all of my questions.